# Rapid Nucleic Acid Extraction for Aquatic Animal DNA Virus Determination Using Chelex 100 Resin via Conventional PCR and Digital Droplet PCR Detection

**DOI:** 10.3390/ani12151999

**Published:** 2022-08-08

**Authors:** Xi Hu, Nan Jiang, Yiqun Li, Yong Zhou, Yuding Fan, Mingyang Xue, Lingbing Zeng, Wenzhi Liu, Yan Meng

**Affiliations:** 1National Demonstration Center for Experimental Fisheries Science Education, Shanghai Ocean University, Shanghai 201306, China; 2Yangtze River Fisheries Research Institute, Chinese Academy of Fishery Sciences, Wuhan 430223, China

**Keywords:** Chelex 100 resin, deoxyribonucleic acid preparation, aquatic animal virus, detection

## Abstract

**Simple Summary:**

Convenient, fast, and high-quality nucleic acid extraction methods are urgently needed in molecular diagnostic testing for viral pathogens in aquaculture. We developed a viral DNA extraction method from diseased tissues and cells using the Chelex 100 resin solution workflow. The only extraction reagents required are the Chelex 100 resin and phosphate-buffered saline. The whole extraction process only takes about 15 min from the tissue homogenate to obtain the DNA. The concentration of extracted DNA is at least 100 ng/µL. This methodology has clear benefits in terms of cost and time saving compared to the commercial kit extraction for aquatic animal DNA virus determination by PCR in the laboratory. In addition, the simplified method using Chelex 100 resin with a pH value of 10–11 presented excellent results in PCR application and could be a standard for the DNA extraction for DNA virus testing in the future.

**Abstract:**

Molecular diagnostic testing for viral pathogens is crucial in aquaculture. The efficient and convenient preparation of pathogenic microbial nucleic acids is the basis of molecular diagnosis. Here, we developed a simplified deoxyribonucleic acid (DNA) extraction method from aquatic animal DNA viruses using the Chelex 100 resin. The nucleic acid was extracted from infected tissues and cell culture for the detection of three common aquatic viral pathogens (CEV, CyHV-2, and GSIV). We compared the extraction effects of a current commercial kit extraction method and the Chelex 100 resin extraction method according to nucleic acid concentration, conventional polymerase chain reaction (PCR), and digital droplet PCR (ddPCR). The results indicated that both extraction procedures could obtain high-quality nucleotide samples. Extracting DNA using the Chelex 100 resin led to better detective efficiency for ddPCR molecular diagnostic testing. The whole process took less than 20 min, and only Chelex 100 resin solution was added to the tissues or cells without multiple tubes being transferred several times. The extracted DNA concentration and the detection sensitivity were high. These results indicated that the Chelex 100 resin solution has the advantages of speed, efficiency, and economy compared to the commercial kit. In addition, the higher pH value (10–11) of the Chelex 100 resin solution markedly improved the detection sensitivity compared to a lower pH value (9–10). In conclusion, the comparison of the Chelex 100 Resin and commercial viral DNA extraction kits revealed the good performance of the Chelex 100 resin solution at pH 10–11 in DNA extraction for PCR amplification from aquatic animal viral samples of tissues and cells in molecular diagnostic testing. It is both rapid and cost-effective.

## 1. Introduction

The global growth of aquaculture has been accompanied by the emergence of many aquatic animal diseases, which have resulted in pandemics across the world. An analysis of 400 emerging disease events in aquatic animals from 2002 to 2017 revealed that more than half of the emerging diseases were caused by viruses [1,2]. Some of these diseases have significant impacts on production and trade, environmental security, and food safety. In order to prevent or control these diseases, the development of rapid and widespread testing must be accelerated through improvements in clinical diagnostics and testing technology.

Since the polymerase chain reaction (PCR) was discovered, rapid molecular methods have greatly enhanced the capabilities of laboratories to diagnose and characterize infectious disease agents. Furthermore, numerous applications in infectious disease diagnostics have been developed [3]. Accordingly, molecular diagnostics based on PCR has become commonplace, accepted in the replacement of serological and microbiologic methods. Nucleic acid amplification for molecular diagnostics remains the gold standard for virus detection [4]. It relies on detecting the specific DNA or RNA nucleotide sequences isolated from samples containing a pathogen. Molecular diagnostics based on PCR methods has played an important role in the field of infectious pathogen detection [5,6]. The ddPCR method has also been gradually applied in the detection of aquatic pathogens [7,8]. Nucleic acid extraction of both DNA and RNA is the most crucial step in any molecular technique, representing the starting point for downstream experiments in molecular diagnostics [9]. The efficiency and quality of nucleic acid extraction directly affect the accuracy, sensitivity, and overall detection efficiency of PCR diagnosis. At present, different nucleic acid sample preparation techniques and commercial kits are often used to extract nucleic acids in clinical and scientific research applications [10,11,12,13,14]. The separation and purification steps traditionally include utilizing phenol chloroform for extraction and/or ethanol for precipitation [15], along with a high salt concentration [16] and excess proteinase K digestion [17]. Nucleic acid extraction kits mainly include magnetic-bead- and spin-column-based methods [18,19]. The traditional nucleic acid extraction methods face problems such as the use of multiple steps, extensive time consumption, and the requirement of large amounts of samples. Commercial kits are effective but expensive. Therefore, simple, convenient, and efficient nucleic acid extraction reagents and methods are very necessary for the molecular detection of pathogens in large population samples.

Chelex 100 resin is a weak cation exchange medium. It can strongly bind divalent metal ions such as magnesium (Mg^2+^) that are required for the activity of metallo-nucleases. High temperatures can result in the release of DNA into solution, as well as facilitate the binding of Chelex resin to magnesium ions. Magnesium ions serve as cofactors to deoxyribonucleases and aid in their activation [20]. The Chelex 100 resin effectively inhibits nuclease activity in complex samples and stabilizes samples for downstream PCR applications. It has long been used for quick and easy DNA or RNA preparation from many sample types, such as blood [21], insects [22], bacteria [23], and viruses [10,11,24]. Carp edema virus (CEV), Cyprinid herpesvirus 2 (CyHV-2), and Chinese giant salamander iridovirus (GSIV) are three common aquatic viral pathogens that cause great losses to aquaculture. CEV is among the major viruses that infect koi or common carp [25]. CyHV-2 is a fatal contagious pathogen that affects goldfish and crucian carp [26]. GSIV, as a severe pathogenic agent, is endemic in cultured Chinese giant salamanders [27]. These three viruses all have double-stranded DNA genomes. Therefore, the development of a method for rapid, efficient, and convenient DNA extraction from these viruses, as well as their detection, is imperative. In this study, we explored the use of the Chelex 100 resin method for the extraction of viral DNA for downstream PCR detection.

## 2. Materials and Methods

### 2.1. Samples

Ten *Cyprinus carpio* and *Carassius auratus* were obtained from daily diseased materials. The gills of *Cyprinus carpio* and kidneys of *Carassius auratus* were collected and stored at −80 °C. They were identified as positive samples infected with CEV or CyHV-2. The giant salamander muscle (GSM) cell line was provided by Prof. Zhang from the Institute of Hydrobiology of the Chinese Academy of Sciences and maintained at 20 °C in medium 199 (Hyclone, Logan, UT, USA) supplemented with 10% fetal bovine serum (FBS). Chinese giant salamander iridovirus (GSIV) was isolated by our laboratory [27]. All animal handling and experimental procedures were approved by the Animal Care and Use Committee of the Yangtze River Fisheries Research Institute, Chinese Academy of Fishery Sciences.

### 2.2. Nucleic Acid Extraction

The steps to extract DNA using Chelex 100 resin were as follows: (1) 5% (*w*/*v*) Chelex 100 resin solution was prepared by dissolving 5 g of Chelex 100 resin (Bio-Rad, Hercules, CA, USA) in 100 mL of ultrapure water before autoclave sterilization and storage at room temperature; (2) the tissue (6 mg tissue sample) was homogenized with 200 µL of 5% Chelex 100 resin solution in a 1.5 mL centrifuge tube and spun for 90 s; (3) the tissue homogenate was transferred into a 1.5 mL centrifuge tube and spun for 5 s; (4) the centrifuged sample was placed in a thermostatic water bath and boiled for 10 min; (5) the sample was cooled to room temperature and centrifuged at 13,000 rpm for 2 min, before extracting the supernatant (i.e., the DNA). To extract the DNA using the viral DNA extraction kit (OMEGA, Norcross, GA, USA), 6 mg of sample was homogenized in 300 µL of phosphate buffer solution (Hyclone, Logan, UT, USA) in a 1.5 mL centrifuge tube, before extracting the DNA according to the manufacturer’s instructions. GSIV was propagated in GSM cells similarly to the method previously described in epithelioma papulosum cyprinid (EPC) cells [27]. The same volume of GSM cell pellet suspension infected by GSIV was used for DNA extraction using the two methods presented above. The concentration and purity of the DNA extracted using the two methods was examined by measuring the 260/280 absorbance ratio using Nano Drop One (Thermo Scientific, Waltham, MA, USA). The DNA was stored as PCR templates at −20 °C.

### 2.3. Optimized Protocol for Chelex 100 Resin Extraction

In order to obtain a better DNA extraction effect using the Chelex 100 resin method, we compared the extracted DNA at pH 9–10 and pH 10–11. First, the Chelex 100 resin solution was adjusted using hydrochloric acid to the appropriate pH (9–10 or 10–11) as monitored by an acidometer (Mettler-Toledo, Shanghai, China). Then, the above-described DNA extraction procedures were followed. Finally, the extracted DNA was measured and preserved at −20 °C.

### 2.4. Quality Assessment by PCR and ddPCR

The pathogenic viruses CyHV-2, CEV, and GSIV were used to evaluate the DNA extraction effects using the two methods. The PCR and ddPCR amplification techniques were performed for viral detection. Some of the detection methods referenced previous publications, while others were established by this study. The primers were designed using the NCBI online primer tool. All primers used in this study are listed in Table 1.

#### 2.4.1. PCR Application

The PCR detection procedure for CEV referenced the standard SC/T 7229-2019 [28]. It was performed with two sets of nested primers, which amplified the 528 bp and 478 bp fragments of the *P4a* gene of the CEV. The PCR detection procedure for CyHV-2 referenced previously published work by Waltzek et al. [29]. The primers of GSIV were designed according to the major capsid protein (MCP) gene sequence (JN099267) using Primer 6.01 software (Version X, La Jolla, CA, USA) (shown in Table 1). The 25 µL PCR reaction mixture consisted of 2.5 µL of 10× Ex Taq Buffer, 2 µL of 10 µM dNTPs, 0.25 µL of Taq DNA polymerase, 1.0 µL each of 10 µM primers, 1 µL of template DNA, and 17.25 µL of H_2_O. PCR reactions were programmed for an initial denaturation step at 95 °C for 4 min, followed by 35 cycles of denaturation at 94 °C for 1 min, annealing at 56 °C for 30 s, and extension at 72 °C for 45 s, with a final extension at 72 °C for 10 min. The PCR products were detected by 1.5% agarose gel electrophoresis.

**Table 1 animals-12-01999-t001:** Viruses and amplification information used in this paper.

Virus	Method	Primer	Sequence (5′–3′)	Size (bp)
CEV	PCR [28]	CEV-P4a-BF	ATGGAGTATCCAAAGTACTTAG	528
		CEV-P4a-BR	CTCTTCACTATTGTGACTTTG	
		CEV-P4a-IF	GTTATCAATGAAATTTGTGTATTG	478
		CEV-P4a-IR	TAGCAAAGTACTACCTCATCC	
	ddPCR [30]	ddCEV-p4a-F	GAAACATGTTTTAGWGTTTTGTAKATTGT	
		ddCEV-p4a-R	CTTGCTCTAGTTCTAGGATTGTATGATG	
		Probe	FAM-CAAGAAACAAACTCTCTTTACTG-MGB	
CyHV-2	PCR [29]	CyHV-2Hel-F	GGACTTGCGAAGAGTTTGATTTCTAC	366
		CyHV-2Hel-R	CCATAGTCACCATCGTCTCATC	
	ddPCR	CyHV-2-F	AGTGTTTGAAGGCTGTCTGGG	
		CyHV-2-R	ACACATTAACCATAGTCACCATCG	
		Probe	FAM-TCAGTACAACCCGTCATGGTACGCC-TAMARA	
GSIV	PCR	GSIV-F	CGTCCAGGTATGCCGTGTTA	320
		GSIV-R	CAATGTACGGGGGTTCGGAT	
	ddPCR [31]	MCP-ddPCR-F	GCGGTTCTCACACGCAGTC	
		MCP-ddPCR-R	ACGGGAGTGACGCAGGTGT	

#### 2.4.2. Digital Droplet PCR Amplification

The ddPCR detection procedures for CEV and GSIV referenced the published works by Wang [30] and Liu [31], respectively. Their primers are shown in Table 1. The ddPCR amplification of CEV and CyHV-2 used the probe procedure. The ddPCR detection of CyHV-2 was performed using the primers of CyHV-2-F/R and a probe (shown in Table 1) designed by Primer 6.0 software on the basis of the DNA helicase gene (EU349287). The assay was performed in a 20 μL reaction volume using the ddPCR™ Supermix for Probes (No dUTP) (Bio-Rad, Hercules, CA, USA). The reaction mixture comprised 10 μL of ddPCR Supermix, 2 μL of each primer (10 μM), 0.5 μL of probe (10 μM), 2 μL of diluted DNA sample, and 3.5 μL of sterile H_2_O. Twenty microliters of each reaction mix was converted into droplets using the QX200 droplet generator (Bio-Rad). Droplet-partitioned samples were then transferred to a 96-well plate. The amplifications were performed with an initial step at 95 °C for 10 min, followed by 40 cycles at 94 °C for 30 s, 60 °C for 60 s, and 98 °C for 10 min, before storage at 4 °C.

The ddPCR amplification of GSIV used the SYBR Green procedure. The reaction mixture consisted of 2 µL of diluted cDNA sample, 10 µL of 2× QX200 ddPCR EvaGreen Supermix (Bio-Rad, Hercules, CA, USA), 0.2 µL of each primer (10 M), and 7.6 µL of H_2_O. The ddPCR cycle profile included one cycle at 95 °C for 5 min, then 40 cycles at 95 °C for 30 s and 55 °C for 1 min, and finally one cycle at 4 °C for 5 min and 90 °C for 5 min. After the amplification, the droplets from each well of the plate were read individually by the QX200™ Droplet Reader, and the data were analyzed using QuantaSoft™ system (Bio-Rad, Hercules, CA, USA). The threshold between positive and negative droplet populations was set manually using per-plate positive and no-template controls as a guide. Reactions with more than 10,000 accepted droplets per well were used for analysis. The absolute initial copy number of target nucleic acid molecules within each sample was approximated by calculating the ratio of positive to total droplets using Poisson statistics [32].

### 2.5. Statistical Analysis

All data are expressed as the mean ± SD (standard deviation), calculated using GraphPad Prism 6.01 software (Version X, La Jolla, CA, USA) through one-way analysis of variance (ANOVA) followed by the Dunnett test. A *p*-value of <0.01 was considered statistically significant. All tests were performed in at least triplicate.

## 3. Results

### 3.1. DNA Extraction

The DNA was successfully extracted using Chelex 100 resin and the viral DNA kit from diseased fish tissues and virus-infected cells. Their procedures and costs are compared in Table 2, indicating a lower cost and higher speed of the Chelex 100 resin.

All results of the DNA extraction are shown in Table 3. It can be seen that both methods showed a great DNA extraction effect according to the concentration. The OD value of A260/A280 was between 1.54 and 2.14 for Chelex 100 resin extraction, while the OD values of A260/A280 were 1.87, 1.98, and 1.93 when extracted by each pathogen kit. The OD values of A260/A230 displayed a large difference between the Chelex 100 resin and viral DNA kit, but the values were similar for the two pH values of Chelex 100 resin. The concentration of DNA extracted from the Chelex 100 resin was higher than that of the DNA extracted from the viral DNA kit, but the quality was not as pure. The different pH values of the Chelex 100 resin solution showed great differences. A pH of 10–11 led to a higher concentration of extracted DNA, but a lower value of A260/A280. The higher pH value of 10–11 could improve the quantity of extracted DNA but did not enhance its purity.

### 3.2. Extraction Effect of Different Methods Detected by PCR

Three viruses (CEV, CyHV-2, and GSIV) were detected using the PCR method. All samples showed concordant results between the Chelex 100 resin method and the commercial viral DNA extraction kit (Figure 1). For CEV, in the second-round amplification, the 478 bp target segment can be clearly observed in Figure 1. The 366 bp fragment of the DNA helicase gene was successfully amplified for CyHV-2 (Figure 1). For GSIV, the 320 bp partial sequence of the *MCP* gene was detected after amplification (Figure 1).

### 3.3. Comparison of Nucleic Acid Extraction Methods by ddPCR

After the nucleic acid extraction using the Chelex 100 resin and viral DNA extraction kit, their individual performance regarding the yield of viral nucleic acids was compared by ddPCR. All three viruses were detectable after nucleic acid preparation, indicating a detectable amount of viral nucleic acid in the respective samples. The copy values of CEV, CyHV-2, and GSIV are shown in Table 4. In contrast to GSIV, isolated from the cell culture specimen with a relatively low gene copy number (43.6 to 116.3 copies/cell), CEV and CyHV-2 both presented a very large gene copy number in their tissues. For CEV, the highest DNA concentration was found in the gill samples of *Cyprinus carpio* extracted using the Chelex 100 resin (pH 10–11), i.e., 5.43 × 10^5^ copies/mg. Chelex 100 resin (pH 9–10) led to the extraction of 2.24 × 10^5^ copies/mg, while the viral DNA kit only led to the extraction of 0.94 × 10^5^ copies/mg. For CyHV-2, the highest DNA concentration was found in the kidney samples of *Carassius auratus* extracted using the Chelex 100 resin (pH 10–11), i.e., 2.64 × 10^8^ copies/mg. Chelex 100 resin (pH 9–10) led to the extraction of 1.65 × 10^8^ copies/mg, while the viral DNA kit only led to the extraction of 0.94 × 10^8^ copies/mg.

The data are plotted in Figure 2. Overall, the viral load of all three viruses showed the following pattern: Chelex 100 resin (pH 10–11) > Chelex 100 resin (pH 9–10) > viral DNA kit. There were significant statistical differences among the groups (*p* < 0.01).

## 4. Discussion

Molecular detection based on PCR amplification is gaining more interest. A crucial step in detecting viral pathogens from clinical specimens is the efficient extraction of viral nucleic acids. The commercially available extraction kits often have many steps for nucleic acid separation, with the sample being transferred many times during the process [14]. These complicated steps can easily result in DNA loss or contamination [33]. In the procedure of viral DNA extraction from tissues using Chelex 100 resin, it only took five steps to homogenize the tissue and transfer it into the centrifuge before spinning, boiling, and centrifuging. The whole process took less than 20 min, and only Chelex 100 resin was added to the tissues. In contrast, DNA extraction using the commercial kit took about 1 h. Furthermore, extraction using Chelex 100 resin did not use organic solvents or require multiple tube transfers and waste liquid collection. Therefore, compared to the viral DNA extraction kit, the procedure using Chelex 100 resin is simple, rapid, and safe.

The method of nucleic acid extraction using Chelex 100 resin has been developed for a long time [34], and it is currently used in many laboratories. Herein, the DNA of three aquatic viruses was extracted using Chelex 100 resin and a commercial kit, and their performance in PCR detection was compared. Conventional PCR assays for the quality of extracted viral DNA showed that the simplified Chelex 100 resin method produced similar results to the viral DNA kit protocol. Further analysis by ddPCR found that the Chelex 100 resin led to higher copy numbers of virus than the detection kit. A previous study presented similar conclusions that the Chelex 100 resin could result in better DNA extraction for PCR amplification in various specimens using different methods [21,35,36]. The ultimate target numbers delivered to a PCR reaction depend on the DNA extraction efficiency, concentration of the extract, amount of extract used in the reaction, inhibition, and the ability to overcome inhibition [33]. In general, The A260/A280 and A260/A230 ratios are used to assess the purity of nucleic acids. The DNA value of A260/A280 is generally greater than 1.8. A value less than 1.8 indicates the presence of protein or phenolic contamination in the extracted DNA. A value of A260/A230 greater than 2.0 indicates a high purity of nucleic acid. A value less than 2.0 indicates the presence of contamination with carbohydrates (sugars), salts, or organic solvents. According to the results in Table 3, it was found that DNA extraction using Chelex 100 resin could lead to a better quantity rather than quality of DNA compared to the commercial kit. Chelex 100 resin, as a chelate-type ion-exchange resin, can bind to a variety of metal ions, which can prevent them from degrading DNA under high-temperature and low-ionic-strength conditions. Moreover, Chelex 100 resin can be removed by centrifugation without affecting the Mg^2+^ concentration in the PCR stage [37]. DNA is also protected by Chelex 100 at high temperatures, while polymerase inhibitors are inactivated during Chelex 100 extraction [35]. The column-based viral DNA extraction kit could yield pure, double-stranded DNA, but it may be subject to significant DNA loss. The Chelex 100 resin extraction method could produce unpurified DNA samples with single-stranded DNA, but could increase the recovery during amplification [38,39]. These results confirm that the Chelex 100 resin method is very advantageous for DNA extraction and can be used in various PCR-based molecular detection approaches.

Chelex 100 resin can be used to extract RNA from pathogens as described above [10,11]. Thus, it is very valuable to establish a method for simultaneously extracting RNA and DNA using Chelex 100 resin. The grass carp reovirus (GCRV) and spring viraemia of carp virus (SVCV) are two RNA viruses that have received great attention for Chinese freshwater aquaculture due to their strong infectivity and high fatality rate. We extracted RNA from grass carp reovirus (GCRV) and spring viraemia of carp virus (SVCV) using Chelex 100 resin at the beginning of this study. However, compared with DNA extraction, RNA extraction using Chelex 100 resin was more difficult, despite some successful cases of RNA extraction using Chelex 100 resin [10,11]. We optimized the RNA extraction conditions using Chelex 100 resin, but the extraction effect was still not particularly satisfactory. On the other hand, RNA extraction with Trizol, the traditional method of RNA extraction, is relatively convenient. Therefore, we believe that Chelex 100 resin is more suitable for extracting viral agents whose genetic material is DNA. In order to obtain a superior DNA extraction effect using Chelex 100 resin, the optimal extraction conditions were studied. No significant improvement was reported when SDS, NP40, T20, and Triton TM X-100 were added to Chelex 100 during DNA extraction from bacterial, viral, or blood samples [21]. Nucleic acids were sensitive to the pH value during the extraction process [40]. Chelex 100 resin is attractive for the extraction of nucleic acid as it can cause cell membrane rupture, as well as DNA denaturation and release, in an alkaline and high-temperature environment [35,37,41]. Therefore, the pH value of Chelex 100 resin solution and the temperature are considered important variables affecting DNA extraction. In the present study, we compared the extraction effect of the Chelex 100 resin solution at pH values 9–10 and 10–11 throughout the extraction process. The results revealed that a pH of 10–11 significantly improved the viral DNA release as reflected by the DNA concentration and ddPCR detection compared to a pH of 9–10. We attributed this to the more alkaline solution of Chelex 100 resin causing greater cell rupture and the release of more virions.

Molecular testing for aquatic animal viral pathogens is crucial to combat the current disease pandemics, especially where there is no cure or vaccine in aquaculture. A number of different PCR assays have been developed and used in pathogen diagnosis. The first step of PCR amplification is to extract nucleic acids from different biological samples. We described a simplified protocol of a high-concentration viral DNA extraction using Chelex 100 resin from diseased tissues and cells. The extracted DNA was suitable for the PCR molecular detection of three common aquatic viruses (CEV, CyHV-2, and GSIV). In terms of the extraction DNA efficiency, Chelex 100 resin was superior to the commercial kit according to PCR and ddPCR assays. Thus, it is well suited for the rapid molecular diagnosis of viruses because of the increased viral DNA yield and its rapid and cost-effective procedure.

We developed a rapid and simplified method for aquatic animal viral DNA extraction using Chelex 100 resin in this study. The concentration and purity of DNA extracted using the Chelex 100 resin and viral extraction DNA kit were evaluated by conventional PCR and digital droplet PCR. The results demonstrated that the Chelex 100 resin was rapid, efficient, and convenient compared to the commercial kit in amplifying specific gene fragments of viruses for pathogen detection.

## 5. Conclusions

In the present study, we reported a method for the extraction of viral DNA from diseased tissue or cell samples of aquatic animals using Chelex 100 resin. The only extraction reagents required are Chelex 100 resin and PBS buffer. The whole extraction process from the tissue homogenate only takes about 15 min to obtain the DNA, with a concentration of at least 100 ng/µL. The Chelex 100 resin solution at pH 10–11 performed well for PCR amplification in the molecular diagnostic testing of aquatic animal viral samples. In addition, we compared the DNA extraction effects of Chelex 100 resin and commercial viral DNA extraction kits in PCR and ddPCR assays. In conclusion, DNA extraction using Chelex 100 resin is rapid, cost-effective, and suitable for DNA preparation for virus detection by PCR.

## Figures and Tables

**Figure 1 animals-12-01999-f001:**
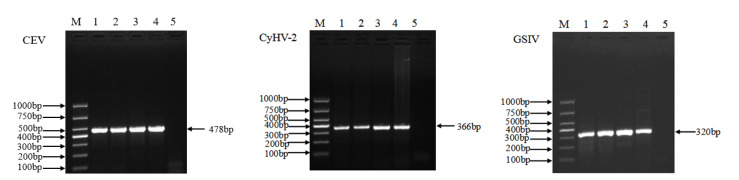
Viral DNA detection of CEV, CyHV-2, and GSIV using Chelex 100 resin and viral DNA extraction kit based on PCR amplification. M: molecular weight marker; lanes 1–5: DNA extracted by Chelex 100 resin (pH 9–10), DNA extracted by Chelex 100 resin (pH 10–11), and DNA extracted by viral extraction kit, positive control, and negative control, respectively.

**Figure 2 animals-12-01999-f002:**
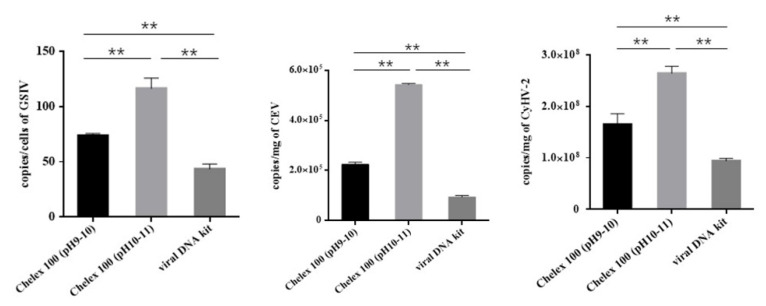
The copy numbers of CEV, CyHV-2, and GSIV obtained from the ddPCR according to the assessment of different DNA extraction methods. Asterisks represent a significant difference.

**Table 2 animals-12-01999-t002:** Comparison of the extraction DNA procedures of the two methods.

Chelex 100 Resin Procedure		Viral DNA Extraction Kit	
Steps	Reagents	Steps	Reagents
1. Prepare 5% (*w*/*v*) Chelex 100 resin solution by dissolving 5 g of Chelex 100 resin in 95 mL of ultrapure water, before autoclave sterilization and storage at room temperature.	Chelex 100 resin; ultrapure water; hydrochloric acid	1. Add 6 mg of tissue to a 1.5 mL centrifuge tube with 250 µL of PBS. Then, homogenize the tissue for 90 s.	PBS
2. Homogenize 6 mg of tissue in 200 µL of 5% Chelex 100 solution in a 1.5 mL centrifuge tube before spinning for 90 s.		2. Add 10 µL of OB Protease and 250 µL of Buffer BL. Add 4 µL of linear acrylamide to 250 µL of Buffer BL. Vortex at maximum speed for 15 s to mix thoroughly.	OB Protease; Buffer BL; linear acrylamide
3. Place 1.5 mL centrifuge tube with sample in thermostatic water bath and boil for 10 min.		3. Incubate sample at 65 °C for 10 min. Briefly vortex the tube once during incubation.	
4. Leave sample to cool to room temperature and then centrifuge at 13,000 rpm for 2 min, before extracting supernatant (i.e., DNA).		4. Add 260 µL of absolute ethanol (room temperature) to lysate the sample, and then vortex at maximum speed for 20 s to mix thoroughly. Briefly centrifuge the tube to collect any drops from the inside of the lid.	Absolute ethanol
		5. Assemble HiBind DNA Mini column in a 2 mL collection tube. Transfer the lysate from step 5 into the column, and centrifuge at 8000× *g* for 1 min to bind DNA. Discard the collection tube and flow-through liquid.	
		6. Place the column into a second 2 mL tube and wash by pipetting 500 µL of HBC Buffer.	
		7. Place the column into the same 2 mL tube from step 6 and wash by pipetting 700 µL of DNA Wash Buffer diluted with ethanol. Centrifuge at 8000× *g* for 1 min.	DNA Wash Buffer
		8. Using a new collection tube, wash the column with a second 700 µL of DNA Wash Buffer and centrifuge as described above. Discard the flow-through and reuse the collection tube for the next step.	DNA Wash Buffer
		9. Place the empty column into the same 2 mL collection tube form step 8, and centrifuge at maximum speed (15,000× *g*) for 2 min to dry the column.	
		10. Place the column into a sterile 1.5 mL microfuge tube and add 50–100 µL of preheated (65 °C) Elution Buffer. Allow tubes to sit for 5 min at room temperature.	Elution Buffer
		11. To elute DNA from the column, centrifuge at 8000× *g* for 1 min. Discard column and store the eluted DNA at −20 °C.	

**Table 3 animals-12-01999-t003:** Concentration and purity of extracted DNA.

Virus	Sample Source	Method	Concentration (ng/µL)	A260/A280	A260/A230
CEV	Gill of *Cyprinus carpio*	Chelex 100 (pH 9–10)	383.9	1.72	0.52
		Chelex 100 (pH 10–11)	526.5	1.70	0.51
		Viral DNA kit	268.2	1.87	2.00
CyHV-2	Kidney of *Carassius auratus*	Chelex 100 (pH 9–10)	231.7	1.61	0.49
		Chelex 100 (pH 10–11)	512.1	1.54	0.53
		Viral DNA kit	285.9	1.98	1.90
GSIV	Affected GSMcell	Chelex 100 (pH 9–10)	115.4	2.09	0.96
		Chelex 100 (pH 10–11)	158.4	2.14	1.04
		Viral DNA kit	40.0	1.93	2.15

**Table 4 animals-12-01999-t004:** Copy number for virus detection using different methods in ddPCR assay.

Virus	Methods	Copy Number (Copies/mg Cell)	SD (Copies/mg Cell)
CEV	Chelex 100 (pH 9–10)	2.24 × 10^5^	7527
	Chelex 100 (pH 10–11)	5.43 × 10^5^	3742
	Viral DNA kit	0.94 × 10^5^	4546
CyHV-2	Chelex 100 (pH 9–10)	1.65 × 10^8^	1.67 × 10^7^
	Chelex 100 (pH 10–11)	2.64 × 10^8^	1.12 × 10^7^
	Viral DNA kit	0.94 × 10^8^	0.41 × 10^7^
GSIV	Chelex 100 (pH 9–10)	73.74	1.6
	Chelex 100 (pH 10–11)	116.3	7. 8
	Viral DNA kit	43. 6	3.6

All mean values are copy numbers derived from each plate in three independent experiments.

## Data Availability

The data supporting the findings of this study are available within the article.

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
