# Peer review of "Rapid Nucleic Acid Extraction for Aquatic Animal DNA Virus Determination Using Chelex 100 Resin via Conventional PCR and Digital Droplet PCR Detection"

_animals, 2022, doi:10.3390/ani12151999_

Round 1

Reviewer 1 Report

Review of ‘’Rapid nucleic acid preparation for aquatic DNA viruses determination using Chelex 100 resin by conventional PCR and digital droplet PCR detection.’’

The present manuscript represents a research study to compare che use of Chelex 100 resin for DNA extraction at different PH to a standard commercial DNA extraction kit. PCR and ddPCR techniques were performed to evaluate the presence of CEV, CyHV-2 and GSIV in different diseased tissues and giant salamander muscle cell line.

Technical methodology (M&M section) seems sufficient to the objectives proposed but introduction, discussion and eventually conclusion sections need ample revision. Conclusion seems ambitious and not supported by evidence especially when quality of DNA, costs and timing of the extraction are described.

The manuscript requires substantial improvement of English language through the text both from a grammatical point of view and in the use of appropriate scientific phrasing and terminology.

In the reviewer’s opinion the manuscript might be reconsidered for approval after major revisions.

Please see below specific comments with line references.

·       Title: we recommend to use the use the term ‘’extraction’’ instead of ‘’preparation’’. Also, we recommend to use ‘’aquatic animals viruses’’ instead of ‘’aquatic viruses’’.

·       corresponding author name and surname is not mentioned.

·       Line 12: Please remove ‘’The’’. Convenient and economical have similar meaning so ‘’economical’’ seems redundant.

·       Line 14: we develop “a viral” not “the viral”, ‘’tissues and cells’’ instead of ‘’tissues or cells’’

·       Line 15: ‘’time saving and cost’’ as benefits needs to be supported by data in the manuscript.

·       Line 16: ‘’simplified referred to resin’’ but should refer to method.

·       Line 17: delete “test”

·       Lines 22-23: the sentence has no meaning or no verb.

·       Line 24: “a current” not “the current”.

·       Lines 27-28: “Extraction DNA….” Should be amended with “Extracting DNA” or “Extraction of DNA….”. ‘’Based ddPCR’’ should be amended with ‘’based on ddPCR’’ ….

·       Line 29: this sentence is not fully supported by data in the text.

·       Line 31: amend ‘’evaluation’’ with ‘’comparison’’

·       Lines 32-34: Please rephrase these two sentences. What is the meaning of ready DNA preparation?

·       Line 41 and 42: not all these diseases have impacts on the aspects you mentioned.

·       Line 43: the term ‘’people’’ is not appropriate.

·       Line 50: the term augmenting is not clear.

·       Lines 53-55: this sentence is redundant. Here we propose to mention ddPCR technique and cite most recent application of this technique in aquatic diseases

- Orioles, M., Bulfoni, M., Saccà, E., Cesselli, D., Schmidt, J. G., and Galeotti, M. (2022). Development and application of a sensitive droplet digital PCR for the detection of red mark syndrome infection in rainbow trout (Oncorhynchus mykiss). Aquaculture 551:737910. doi: 10.1016/j.aquaculture.2022.737910;

- Wang et al., 2021 –already in your bibliography list).

·       Line 67: to use convenient and economical please refer to previous comments. This will not be amended again through revision.

·       Line 71: si intede “the use of high temperatures in the extraction process”? (frase troppo buttata lì).

·       Line 77: all the viruses acronyms must be written here.

·       Line 80: the sentence cannot start with ‘’And’’.

·       Line 84: What is the meaning of ‘’practical’’ here?

·       Line 89: Please write the exact number of samples Va necessariamente precisato il numero di campioni esaminati. Inoltre qui andrebbe specificato la Sample Source che poi è illustrata in tabella 2 (non basta dire tissues).

·       Line 90: non credo si possa inserire il nome di una persona in un articolo scientifico, a parte citare gli autori di articoli.

·       Line 95: please insert protocol number of approval.

·       Lines 96-113: please rephrase all this paragraph. It seems it has been copied and pasted from manufacturer instructions, which is not a standard in scientific manuscripts.

·       Line 117: please state clearly what has been used to lower PH here.

·       Line 118: the use reserved does not make sense here.

·       Paragraphs 2.3 and 2.4 : these paragraphs needs grammar checks and english revision.

·       Line 125: andrebbe scritto qui (e non nei risultati) che si tratta di una nested PCR e che il secondo amplicone ha dimensioni di 478 bp.

·       Table 1: what is the probe for ddPCR for GSIV?

·       Lines 141-143: the sum is 18.5 not 20.

·       Lines 159: revise english here. The meaning is not clear. How many samples were analysed in total?

·       Line 161: The title of the paragraph should be ‘’DNA extraction”.

·       Lines 163-165 e 166-167: These details are part of Material and Methods.

·       Line 165: We would recommend to report results from A260/A230 analysis to verify the presence of possible contamination.

·       Line 168: meaning here is not clear. Please specify and rephrase.

·       Line 174: This is not correct based on your results. 1.70 is less than 1.72 and1.54 is less than 1.61. The quantity extracted improves but not the quality.

·       Line 176: Table 2, caption/title: is better to write: “Concentration and purity of extracted DNA”.

·       Line 177: we recommend to write: “Extraction effect of different methods detected by PCR”.

·       Line 178: please rephrase.

·       Lines 181-182: it would be best to write ‘’nested’’ primers. The gene amplified should written in M&M section.

·       Figure 1, caption: it is best to refer to pH 9-10 e pH 10-11, and not only to 9-10 e 10-11. Also write “Lane 1-5”, and not only “1-5”:

·       Line 206: per GSIV, copy n./mg o per cell?

·       Table 3: Copy number/mg refers to the tissue? Was the number of cells in 250 ul determined?Please double check standard deviation for CyHV-2.

·       Line 216: Please amend the term ‘’graph the data’’

Discussion: this section need extensive english grammar revision.

Line 242: “by” non “of Chelax 100 …”.

Line 245: The adjectives rapid efficient and convenient seem to be related to the commercial kit. Please rephrase.

Lines 247-248: redudant.

Lines 254-255: what about commecial kit? How long does it take to use these?

Lines 259-260: these sentences have to be rephrased.

Line 261: please refer to extraction methods rather than to DNA extractions.

Line 270: please refer to either efficiency or concentration, not both, because only the concentration is higher, the quality is a little lower.

Line 274: the sentence has to be rephrased.

Line 279: this sentence is too colloquial

Lines 296-297: which diseases you are referring to?

Conclusions: please consider to include them into discussion as they are currently redundant. Rephrasing and english grammar need revision.

Author Response

Dear Reviewer,

Thank you very much for reviewing our manuscript and give us your valuable comments. We revised the manuscript based on all the comments,and try our best to minimize the questions in the manuscript. The attachment is the reply to these questions, please check it.

Thank you and best regards.

Yours sincerely,
Xi Hu

Reviewer 2 Report

The authors present data on the use of Chelex 100 resin for DNA extraction of various aquatic DNA viruses. The efficiency of DNA extraction was analysed depending on the pH value and compared to a commercial standard DNA extraction kit. Positive samples from two organs and one cell culture were used for the comparative studies.

Major comments:

·         The number and variety of the tested positive samples is not enough for the conclusion stated out. More clinical samples per virus and dilution series of positive sample material in negative background matrix must be tested for a statement regarding analytical and diagnostical sensitivity and specificity. Please use the guidelines of the OIE/WOAH for the validation of molecular diagnostic tests as basis for evaluation (Chapter 1.1.6. Principles and methods of validation of diagnostic assays for infectious disease (version adopted in May 2013)

·         Normally, RNA and DNA viruses are analysed diagnostically and it is therefore always advantageous if both viral DNA and viral RNA are extracted from clinical samples. No comments were made on this either in the introduction or in the discussion. The advantage of the commercial standard extraction kit here is certainly the parallel extraction of RNA and DNA. This should be noted. It would be even better to evaluate the newly described Chelex method for the extraction of viral RNA as well.

·         In the text, the lower costs, effort and high speed of the Chelex method are repeatedly cited as advantages. However, this statement is not confirmed with corresponding data. A table should be introduced in which the costs and time required are compared between the methods.

·         Basically, the advantages of the Chelex method are mentioned in particular. Disadvantages, such as insufficient RNA extraction or sub-optimal quality of the extracted DNA, are not clearly named. A detailed comparison of all advantages and disadvantages of the extraction methods is recommended.

·         The English language of the text should be substantially revised. The use of an editing service is strongly advised.

Minor comments:

·         Line 77: the virus names including abbreviations should be described first

·         L 97: What is the standard pH after preparation of the 5% Chelex 100 resin? How was the pH adjusted? Why pH 9-10 and 10-11? In both cases you define pH10 as possibility. Thus, the pH must be defined more exactly.

·         L 101: homogenisation procedure is not described

·         L 103: Place the centrifuge …in water bath…? That is very special!

·         L 109: What are EPC cells?

·         Table 1:

o   References for the oligos should be listed in the table.

o   Size of the real-time PCR product should be also defined in the table.

o   ddPCR of GSIV without FAM-labelled probe? Is it a SybrGreen-procedure? Should be define more detailed.

·         Figure 1: define the sample per lane exactly.

·         L 206: GSIV with a very large copy number?? That is not correct.

Author Response

(The authors gave the same response as above.)

Reviewer 3 Report

The fact that viral DNA extraction using Chelex100 resin was more efficient than that using commercial DNA extraction kits, especially under pH 10-11 conditions, is a good news for the future detection of DNA viruses in trace amounts, and we believe that the scientific value of this paper is well worth the effort.

Author Response

Dear Reviewer,

Thank you for your approval of this article. We will make more  scientific research in the future.

Thank you and best regards.

Yours sincerely,
Xi Hu

Round 2

Reviewer 1 Report

Dear Authors,

Thanks for your revised version of the manuscript. Although we appreciate the effort to amend English inaccurancies and we consider the content valid for publication, we strongly recommend to revise extensively English language in the text, as this is not acceptable for publication in the present form.

The manuscript requires major revisions. We recommend to use a professional proofreading service to meet standard English scientific language.

Please find below specific comments, but please note that English grammar revisioni is needed throughout all the manuscript:

Lines 15-17: avoid to put acronyms in the summary

Lines 17-20: revise English grammar

Line 32: revise English grammar

Lines 33-35: revise English grammar. Never start a sentence with ‘’And’’. Please amend throughout all the text as well (I am not going to revise it specifically anymore)

Line 62: this seems to be redundant and not correct grammatically

Line 100: Please specify the meaning of this sentence.

Line 103: Fish or tissues?

Line 111 – paragraph 2.2: this requires extensive english grammar revision. For example you can write..“The Chelax 100 resin solution was prepared at 5% (W/V) by dissolving, etc., etc…. ‘’

Line 123: what is the specific name of the kit?

Line 127: What volume refers to ‘’same volume’’?

Line 129: concentration and purity rather than quality

Line 136: what is the initial PH?

Line 139: DNA concentration and purity.

Line 2.4.1: PCR application or amplification (like at 2.4.2)?

Line 204: amend ‘’respectively’’ with ‘’for every pathogen’’…

Line 208 – Table 3: the low A260/A230 and A260/A280 are suggesting that there is high contamination. The resin seems to work well (low contamination) only with extraction from cells.

Line 210: this is valid for the two tissues; it is good in the cells

Line 212: the resin can take many contaminants that can influence PCR or ddPCR analysis.

Figure 1 – caption: M: molecular weight markers.

Line 248: Please explain how you determined the number of copies/cells

Line 248: whose tissues are these?

Lines 250-251: revise English grammar

Lines 283-289: this part would go at the end of discussion

Lines 300-302: revise English grammar

Line 320: Not the concentration, but the purity!

Line 321: this seems not true. It is higher than 1.8 only strating from cells.

Line 327: rather than purity, non rather than purify.

Line 375: high concentration rather than high quality

Author Response

Dear Reviewer’s

Thank you for your precious comments and advice.We have tried our best to revise the manuscript according to your comments and suggestions.We used an editing service you recommended to revise the article including some typos, grammatical errors and long sentences, etc. The attachment is the reply to these questions, please check it.

Thank you and best regards.

Yours sincerely,

Xi Hu
